# Possible Mechanisms of Subsequent Neoplasia Development in Childhood Cancer Survivors: A Review

**DOI:** 10.3390/cancers13205064

**Published:** 2021-10-10

**Authors:** Jarmila Kruseova, Ales Vicha, Barbara Feriancikova, Tomas Eckschlager

**Affiliations:** Department of Pediatric Hematology and Oncology, 2nd Faculty of Medicine, Charles University and University Hospital Motol, V Uvalu 84/1, 150 06 Prague 5, Czech Republic; Jarmila.Kruseova@fnmotol.cz (J.K.); Ales.Vicha@fnmotol.cz (A.V.); barbara.feriancikova@gmail.com (B.F.)

**Keywords:** subsequent malignant neoplasms, childhood cancer survivors, genetic factors, cancer predisposition syndromes, chemotherapy, radiotherapy

## Abstract

**Simple Summary:**

Due to improvements in the treatment of childhood cancer, around 80% of children are cured. However, childhood cancer survivors are at risk of developing late effects, including subsequent malignant neoplasm; these are defined as histologically different cancers, which appear after treatment for primary cancer. The risk of subsequent malignant neoplasm formation is influenced mainly by previous anticancer therapy (chemotherapy, radiotherapy, immunotherapy, and targeted therapy), genetic factors, and the length of survival. For these reasons, at present, for the treatment of tumors with a good prognosis, it is important to consider the possible risk of late side effects e.g. use of radiotherapy in Hodgkin’s lymphoma only if chemotherapy does not induce complete remission or in nephroblastoma only in locally advanced stages. Therefore, we study risk factors for the development of subsequent malignant neoplasm. In the review, we present possible risk factors for the development of subsequent neoplasm.

**Abstract:**

Advances in medicine have improved outcomes in children diagnosed with cancer, with overall 5-year survival rates for these children now exceeding 80%. Two-thirds of childhood cancer survivors have at least one late effect of cancer therapy, with one-third having serious or even life-threatening effects. One of the most serious late effects is a development of subsequent malignant neoplasms (histologically different cancers, which appear after the treatment for primary cancer), which occur in about 3–10% of survivors and are associated with high mortality. In cancers with a very good prognosis, subsequent malignant neoplasms significantly affect long-term survival. Therefore, there is an effort to reduce particularly hazardous treatments. This review discusses the importance of individual factors (gender, genetic factors, cytostatic drugs, radiotherapy) in the development of subsequent malignant neoplasms and the possibilities of their prediction and prevention in the future.

## 1. Introduction

Advances in diagnostics, therapy, and supportive care have improved outcomes for children diagnosed with cancer, and overall 5-year survival rates for these children now exceed 80%. One in every 1000 young adults at age 20 is a childhood cancer survivor (CCS) in the USA. Two-thirds of CCSs have at least one chronic or late-occurring complication (late effect) of cancer therapy, with about one-third having serious or even life-threatening complications [1,2,3,4,5]. Late effects can be anticipated based on therapeutic exposures, but multiple factors modify the risk for an individual patient. One of the most serious late effects are subsequent malignant neoplasms (SMNs), defined as histologically distinct malignancies that develop in children with primary cancer. SMNs are the leading cause of non-relapse late mortality. Their incidence in CCSs increases with sustained age, with the cumulative incidence exceeding 20% at 30 years after diagnosis of primary cancer [1,6]. For example, the British Columbia Cancer Registry study found an 18-fold increased risk of death from SMNs, a 3-fold increased risk of death from non-malignant disease, and a 19-fold increase risk of death from circulatory disease in Hodgkin’s lymphoma survivors compared to matched controls [5]. In our group of CCSs treated between 1975 and 2018, 4.3% developed at least one SMN until the year 2019, and the mortality for an SMN was 38.2%, i.e., 1.6% of all survivors died of SMN [7]. Our results and the literature show that the incidence of SMNs is relatively common, occurring in about 3–10% of CCSs and associated with high mortality [4,8,9,10,11,12,13]. The occurrence of SMNs in adolescent and young adult (AYA) cancer survivors was lower than in CCSs, as shown by the results of an extensive US study of AYA cancer survivors [14]. This suggests that children are more susceptible to SMNs, probably due to the greater sensitivity of developing tissues to chemotherapeutic drugs and particularly to the radiation. Differences in the incidence of SMNs in CCSs and AYA cancer survivors are also partly due to differences in the incidence and treatment of first cancers in children and AYA. Leukemia, Hodgkin’s lymphoma, soft-tissue sarcomas, nephroblastoma, bone sarcomas and brain tumors predominate in CCSs [4,5,6,7,8,9,10,11,12,13,14,15,16,17,18,19], while Hodgkin’s lymphoma, melanoma, germ cell tumors, bone sarcomas, leukemia and brain tumors are the most frequent in AYA cancer survivors [20,21]. The most common SMNs in CCSs include breast cancer, thyroid cancer, brain tumors, soft-tissue sarcomas, leukemias and non-melanoma skin cancer (NMSC) [7,8,9,12,15,16,17,18]. The most common SMNs in AYA cancer survivors are breast cancer, soft-tissue sarcomas, lymphomas, leukemias, and thyroid carcinomas [14,20,21]. 

From the above, it is clear that SMNs are the relevant medical, social, and economic problem. This review provides a comprehensive overview of the mechanisms and risk factors for the development of SMNs. 

## 2. Gender

Many studies focusing on CCSs confirmed a higher incidence of SMNs in females than in males [4,15,16,17,18,19]. Moreover, in our group of SMNs, we observed a higher proportion of women than in the group of survivors without SMN, but this difference was not statistically significant [7]. One possible explanation for this difference is the relatively high incidence of subsequent breast cancer, particularly after chest irradiation among women but not men [22]. In a cohort of 1380 childhood Hodgkin’s lymphoma survivors followed by Late Effect Study Group, there were 88 SMNs, where breast cancer was the most common subsequent solid tumor, seen in 17 women (19% of all SMNs). Sixteen of them developed breast cancer within the radiation field, while one underwent neck irradiation [23]. Therefore, one may speculate that this high frequency of subsequent breast cancer in women survivors of Hodgkin’s lymphoma is caused by irradiation of the mammary gland. Forty-five cases (17%) of female breast cancer were found in 261 SMNs in CCSs in the DCOG LATER (Dutch Childhood Oncology Group-Long-Term Effects After Childhood Cancer) cohort. The chest radiotherapy and a higher cumulative dose of doxorubicin were confirmed as risk factors (hazard ratio 1.1 for doxorubicin cumulative dose ≤270 mg/m^2^, 2.6 for dose 271 to 443 mg/m^2^, and 5.8 for >443 mg/m^2^) [9]. Moreover, CCSS proved higher anthracycline dose as a risk factor for subsequent breast cancer development [8]. St. Jude childhood and adolescence study in Hodgkin’s lymphoma survivors reported more common SMNs among females even after excluding subsequent breast cancer [24]. 

These findings could be partially explained by gender-dependent acute toxicity of some cytostatics. The experimental study found out higher IC_50_ of etoposide on male EBV-transformed B-lymphoblastoid cell lines from healthy persons than female ones [25]. Some clinical trials reported that the female gender is a predictor of more serious chemotherapy-induced toxicity (gemcitabine and carboplatin induced pulmonary toxicity, oral mucositis after different chemotherapeutic protocols) [25]. Severe toxicity in Ewing’s sarcoma patients treated in the Euro-EWING99-R1 trial was more common in women than in men, regardless of toxicity type [26]. Hormonal influences may also contribute to this difference, because the prevalence of endocrine late effects had about half of CCSs [27]. Nevertheless, the cause of the higher incidence of SMN in women has not yet been fully elucidated.

## 3. Genetic Syndromes

Although most cancers are sporadic, i.e., their development is multifactorial, up to 10% of cancers are related to a hereditary genetic mutation [28]. So far, over 300 hereditary cancer susceptibility syndromes are known, and about 40 of them are predisposed to childhood cancer [29,30,31]. Table 1 lists the syndromes with increased risk of childhood cancer. Hereditary mutations of some tumor suppressor genes, e.g., *TP53*, *Rb1*, *APC*, *WT1*, that cause loss of their function are responsible for hereditary cancer susceptibility syndromes; however, they contribute only to a small proportion of all cancers [32]. Besides, above-mentioned high-risk hereditary mutations, the genetic background may be involved in the predisposition to cancer. Plenty of polymorphisms or gene variants may change the risk of cancer. Various combinations of those low-risk genes may significantly influence the risk of cancer development [28]. One may speculate that such combinations of polymorphisms or gene variants may participate in SMNs development. These low-risk genes may be studied by genome-wide association studies and by mass, whole-genome sequencing [32]. Thus, it can be concluded that, in a significant number of cancers, genetic factors are more or less involved in their development.

Kingston et al., in their study, analyzed 161 SMNs; 53 of them (33%) had genetic factor with increased risk of cancer development (hereditary retinoblastoma, neurofibromatosis I, Gorlin syndrome, Turcot syndrome, tuberose sclerosis complex, MEN2A, Werner syndrome). Moreover, eight patients with SMNs had a history of cancer in at least one first-degree relative [33]. Participation of genetic influences in SMNs origin proved Meadows et al. in their study. They found an increased risk of cancer in the siblings and particularly offspring of CCSs with SMN compared with the general population. Further, female CCSs with breast cancer in their family history had increased incidence of subsequent breast cancer, and survivors with a family history of skin cancer had more frequently subsequent NMSC [16]. Similar results showed CCS Study focused on NMSC. Subsequent NMSCs were more frequent in survivors of childhood and adolescent cancer with skin cancer in family history than in survivors with family history without skin cancers [34].

Broniscer et al., in their study of CNS tumor survivors, found in 7 of 24 (29%) survivors with subsequent neoplasms (SNs) (malignant, NMSC and meningiomas) genetic predisposing syndrome (Gardnery syndrome, Gorlin syndrome, Li-Fraumeni and neurofibromatosis 2) [36]. In a large cohort of patients with subsequent NMSCs, no genetic syndrome was detected, but they had a significantly higher occurrence of skin cancers in family history than the group of survivors without subsequent NMSC [34].

Hereditary cancer susceptibility syndromes (neurofibromatosis type I, Li-Fraumeni syndrome, *Rb1* mutation, MEN IIb syndrome, familial adenomatous polyposis, Gorlin syndrome, Peutz–Jaghers syndrome, Rubinstein–Taybi syndrome, *NSB1* mutation, and Beckwith–Wiedemann syndrome) were present in 14.7% of CCSs with SMNs in our group [7]. The data may be underestimated, as not all patients with SMNs have been genetically tested. 

St. Jude Lifetime Cohort Study described results of genome sequencing of 3006 CCSs. In 5.8% of survivors, pathogenic or likely pathogenic variants were detected in specific cancer predisposition genes, e.g., *Rb1, NF1, BRCA2*. Survivors with germline variants had an increased risk of SMNs, particularly breast cancer as the second neoplasm [37]. Another study of the St. Jude Lifetime Cohort, that includes 207 survivors who developed subsequent breast cancer and 2774 who had no SMNs, examined genome-wide germline mutations and polymorphisms by Affymetrix Genome-Wide Human SNP Array 6.0. Among CCSs who received at least 10 Gy radiation to the breast region, the locus at 1q41 marked by rs4342822 and locus at 11q23 rs74949440 were associated with a risk of subsequent breast cancer, while locus at 1q32.3 (rs17020562) was risky in survivors with irradiation of this region by dose lower than 10 Gy. Further, they also identified seven other chromosomal regions with associations for subsequent breast cancer risk [38]. The same group estimated variants in one of 156 cancer predisposition genes among 2450 adult CCSs using whole-genome sequencing, with 11.8% of them carrying a pathogenic/likely pathogenic variant, and they frequently detected variants of *Rb1*, *NF1*, and *BRCA2* genes [39].

Large international group studied polymorphisms of mismatch repair *MLH1* gene in subsequent AML and breast cancer in both childhood and adult Hodgkin’s lymphoma survivors treated by methylating agents. MLH1-93 variant allele had higher frequency in therapy-related AML than in de novo AML [40]. The St. Jude late effects group examined whole-genome and whole-exome sequencing in more than four thousand CCSs, in which 11.2% developed SNs and evaluate 127 genes in 6 major DNA repair pathways (Fanconi anemia, homologous recombination, nonhomologous end joining, nucleotide excision repair, mismatch repair, and base excision repair). They identified 538 pathogenic germline mutations in 98 genes in 11.5% of CCSs. Mutations in homologous recombination genes were associated with an increased rate of subsequent breast cancer in females, especially after chest radiotherapy or chemotherapy with anthracycline; moreover, these were also associated with subsequent sarcoma after higher doses of the alkylating agent. Mutations in nucleotide excision repair genes were associated with subsequent thyroid cancer in survivors after neck radiotherapy of at least 30 Gy [41]. Whole-exome sequencing and analysis of 476 genes involved in DNA damage response or radiation sensitivity syndromes of the CCSS cohort (retrospective study of CCSs in USA) was used to identify whether rare variants in those genes affect risk of SNs known to be related to radiotherapy e.g., basal cell carcinoma, breast cancer, meningioma, thyroid cancer, sarcoma. Higher risk of SNs out of irradiation field but not with SNs in irradiated field was associated with homologous recombination repair gene variants. *EXO1* variants (DNA double-strand break repair gene) were more commonly detected in survivors with SNs in the irradiated field [42]. Authors assume that examination of DNA repair genes allows identifying survivors at the increased risk of SMNs and implementing personalized screening.

A molecular study of 650 childhood Hodgkin’s lymphoma survivors who had received radiotherapy studied the association between polymorphisms in *glutathione-S-transferase M1 (GSTM1), glutathione-S-transferase T1 (GSTT1)*, and *XRCC1* and SMNs. GSTM1 and GSTT1 participate in protection against oxidative damage, and XRCC1 is involved in repairing DNA single-strand breaks [43], while polymorphisms of those genes have been associated with several primary cancers [44,45]. CCSs lacking GSTM1 had an increased risk of any SMNs including these within the radiation field. Other associations were not statistically significant (*XRCC1* polymorphisms and increased subsequent breast cancer and lacking either *GSTM1* or *GSTT1* and higher incidence of subsequent thyroid cancer) [43]. Japan Koseisho Leukemia Study Group analyzed gene polymorphisms of *NAD(P)H:quinone oxidoreductase (NQO1), glutathione S-transferase (GST)-M1* and -*T1*, and *CYP3A4* in 58 patients with subsequent myelodysplastic syndrome or acute myeloid leukemia in adult cancer survivors treated by chemotherapy and/or radiotherapy (sMDS), as well as in 411 patients with de novo acute myeloid leukemia (AML); indeed, those enzymes are important for the metabolism of cytostatics. Homozygous *Ser/Ser* genotype of *NQO1* at codon 187, causing loss of function, was more frequent in the patients with sMDS than in those with de novo AML and healthy controls [46]. Felix et al. examined 99 de novo and 30 treatment-related childhood leukemias for the polymorphism in the promoter region of the *CYP3A4* gene, while 19 of 99 de novo and only 1 of 30 treatment-related leukemias had the *CYP3A4-V*. 9 of 42 de novo leukemias had *MLL* gene translocations, and none of 22 treatment-related leukemias with *MLL* gene translocations had *CYP3A4-V*. These results indicate that wild type *CYP3A4-W* genotype may be at increased risk of treatment-related leukemia, since it may produce DNA-damaging reactive drug metabolites [47]. Polymorphisms of RAD51 (*RAD51*-G135C) and *XRCC3* (*XRCC3*-Thr241Met) double-strand break repair genes were increased both in de novo and particularly therapy-induced AML in a study that included both adults and children. The risk of AML is further growing in the persons with *GSTM1* deletion [48]. The above-mentioned facts prove that single polymorphisms usually only have a small effect on cancer risk [49].

There were sequenced *TP53* in samples of subsequent tumors, and peripheral blood in 37 pediatric cancer survivors with SMNs without a family history of cancer predisposition syndrome. Somatic mutations were detected in two samples of subsequent cancers (osteosarcoma and pleomorphic sarcoma), where a patient suffering from pleomorphic sarcoma had a heterozygous germline *TP53* mutation [50]. These findings support the known fact that hereditary mutations of *TP53* are not very common. In contrast, Li-Fraumeni syndrome is characterized by the multiple occurrences of various cancers from an early age, and thus it may be found in some patients suffering from SMNs [51].

A German center study found 51 SMNs in 317 survivors of heritable retinoblastoma. External radiotherapy is the main risk factor for soft-tissue sarcoma and osteosarcoma in the radiation field. The incidence of SMNs was four times higher in survivors heterozygous for an oncogenic *Rb1* variant than with *Rb1* mosaicism [52].

Several other findings indirectly suggest genetic influences in the development of SMNs. The CCSs results found SNs (both malignant and benign) in 9.6% of CCSs, and 27.9% of them developed second SNs. In the authors’ opinion, survivors who develop multiple SMNs, with the exception of those in the irradiated field, are highly suspected of genetic predisposition [53]. A study of Australian Childhood Cancer Registry (ACCR) shows that CCSs who have not received chemotherapy or radiotherapy had a higher risk of cancer than the general population. However, it was lower than that of survivors treated with chemotherapy and/or radiotherapy [12]. Survivors of childhood rhabdomyosarcoma, particularly young children with pleomorphic and embryonal type, are at increased risk of SMNs independently of radiotherapy. In contrast, survivors of other histological types of rhabdomyosarcoma had lower risk, and SMNs in this group were connected with radiotherapy [54]. Therefore, the authors speculate about the genetic influences. CCSs after germ cell tumors and carcinomas had a very high incidence of SMNs (25%) in the Hungarian CCSs study; therefore, the authors considered some predisposition syndromes or polymorphisms [13]. Subsequent breast cancer after higher doxorubicin dose (see above) was more frequent in survivors of “Li-Fraumeni syndrome-associated childhood cancers” (leukemia, brain tumors, and non-Ewing’s sarcoma) than in survivors of other cancers [9]. However, *TP53* mutation testing was not performed in this cohort of survivors; therefore, the considerations about Li-Fraumeni syndrome are only speculative. The participation of genetic factors in the formation of SMN may also be evidenced by the frequent coincidence of malignant and benign SNs. In our group of 170 survivors with SMNs (4.2% of all survivors), 34 had both malignant and benign SN (0.8% of all survivors) [7] and we found coincidence of sporadic renal angiomyolipomas with SNs [55]. Moreover, incidence of sporadic renal angiomyolipomas in CCSs was approximately ten times higher than in the general population [55]

The above-described results of clinical studies demonstrate the role of genetic background in the development of SMNs. So far, we cannot detect all genetic risk factors for the development of SMNs, except for hereditary cancer syndromes, which occur rarely. Therefore, extensive research will still be needed to detect all genetic pathways for the development of SMNs. In contrast, with the introduction of whole-genome sequencing, it can be expected that cancer predisposition syndromes will be diagnosed more frequently, and new genetic factors that influence developing of SMNs will be found [37]. 

## 4. Chemotherapy

Anticancer drugs are carcinogenic, since they destroy both cancer and normal cells. For example, etoposide is associated with an increased risk of acute myeloid leukemia (AML), and alkylating agents and anthracyclines have been associated with many different cancers in CCSs [9]. Moreover, there are also new reports of the occurrence of SNs after targeted therapy in adult cancer survivors; see below.

A number of studies focused on SMNs in CCSs have studied the relationship between administered anticancer drugs and the risk of developing SMNs. Increased doses of alkylating agents, anthracyclines, and epipodophyllotoxins increase the risk of any SMN, as found in Childhood Cancer Survivor Study Cohort [16]. In the DCOG LATER study, doxorubicin was associated with an increased risk of female breast cancer, and cyclophosphamide increased the risk of sarcoma, particularly bone, both in a dose-dependent manner [9]. Survivors treated with high doses of alkylating agents and platinum agents had increased incidence of SMNs in the US and Canadian CCSs studies [19]. Kingston et al. found increased frequency of previous therapy by alkylating agents, especially by cyclophosphamide, in patients with SMNs [33]. Alkylating agents are a risk factor for developing subsequent leukemia in childhood Hodgkin’s lymphoma survivors [23]. We found cyclophosphamide as a risk factor for developing subsequent urinary bladder sarcoma [7]. The above-mentioned studies show that alkylating cytostatics, especially cyclophosphamide and anthracyclines, are particularly hazardous drugs [8,16].

Several studies reported subsequent leukemias, particularly acute myelogenous leukemia (AML) associated with translocations of the *MLL* gene at chromosomal band 11q23 when etoposide or other topoisomerase II inhibitors (topo II i) were used in therapy of primary cancer [56,57,58]. Alkylating agents cause subsequent AML, mostly preceded by myelodysplastic syndrome (MDS), and the interval from the first tumor is usually 5–7 years. Complete or partial deletion of chromosome 5 or 7 is frequent, and blasts are type M1 or M2 according to the FAB classification. The risk of MDS with subsequent AML depends on the cumulative dose of alkylating agents, while epipodophyllotoxins and other topo II i induce leukemias with *MLL* gene translocations at chromosome bands 11q23, t(8;21), t(3;21), inv(16), t(8;16), t(15;17), or t(9;22) [47,56,57]. They occur early after finishing therapy by topo II i, usually within three years, and have FAB M4 or M5 morphology, but there were also described other FAB AML subtypes, MDS, acute lymphoblastic leukemia (ALL), or chronic myelogenous leukemia (CML) [57]. Epipodophyllotoxins (etoposide and teniposide) are the most common cause of secondary AML, but may also occur after therapy with other classes of topo II i (e.g., anthracyclines). 

The relationship between etoposide dose and the incidence of secondary AML is not clear, but it was found that children with ALL treated with the high (4–5 g/m^2^) cumulative dose of etoposide have a high risk of developing secondary AML (cumulative risk 5–12%). In contrast, children treated with lower doses (1.5–2 g/m^2^) of etoposide for germ cell tumors have a lower risk [59]. This would suggest a dependence of subsequent AML on the cumulative dose of etoposide.

A comparison of subsequent leukemias induced by topo II i and alkylating cytostatics is shown in Table 2. The prognosis of alkylating-agent-induced subsequent leukemia is worse than that of de novo leukemias, and the prognosis of topo II i induced leukemia is extremely poor [57,60]. Typical karyotypic abnormalities in topo II i associated AML are balanced translocations. A supposed mechanism is inhibition of the religation step by topo II i, which induces double-strand breaks, and MLL region is sensitive to etoposide-induced cleavage, as shown in in vitro studies [61]. *MLL* translocations in topo i induced-AMLs are caused by changes in chromatin structure and cryptic promoter activity. The *MLL* breakpoint cluster region (bcr) contains areas of hypersensitivity to DNase I, cryptic promoter activity and region for binding of transcription factor CTCF. Increased risk of translocations is caused by increased chromatin accessibility induced by DNase I and/or cryptic promoter activity. Moreover, regions hypersensitive to DNase I were described additionally in other bcr regions of genes that are rearranged in translocations in topo i induced-AMLs - *AF9*, *AF4*, *AML1*/*RUNX1* and *ETO* [61].

Another mechanism involved in the development of SMNs in CCSs is anticancer treatment-induced premature aging. Cells derived from multicellular organisms have finite replicative potential in a cell culture. The link between replicative senescence and human aging has been proved by finding of the accumulation of senescent cells with advancing age, correlations between in vitro replicative potential and donor age, and the diminished in vitro replicative potential of cells from individuals with premature aging syndromes. Analysis of telomere lengths in cells derived from peripheral blood of humans over the age of 60 revealed that individuals possessing shorter telomeres than age-matched controls had significantly poorer survival rates [62]. Senescent cells accumulate with normal aging and contribute to age-related pathologies by inhibiting tissue regenerative capacities. It was described that chemotherapy (doxorubicin, cyclophosphamide, hydroxyurea) induces cellular senescence [63,64], and it is supposed that this phenomenon may participate in the occurrence of SMNs. Several studies suggested an increased risk of cancer in those with the shortest telomeres in blood leukocytes and buccal mucosae cells compared to the longest ones [65,66]. One explanation for this phenomenon is decreased chromosomal stability caused by telomeres shortening. Another study found an increased risk of colorectal cancer in people with shortened and contrary lengthened telomeres in peripheral blood leukocytes [67].

Adult breast cancer survivors treated by chemotherapy have increased expression of markers of cellular senescence (*p16INK4a, ARF*) in T lymphocytes and increased levels of senescence-associated cytokines (*VEGFA* and *MCP1*) in serum, comparable with the effects of 10 to 15 years of chronologic aging in independent cohorts of healthy donors [68]. Accelerated aging in CCSs is suggested because of a high prevalence of frailty among young adult CCSs, similar to that of adults 65 years and older [69].

The above-mentioned studies [65,66] found an increased risk of cancer in persons with shorter telomeres in blood leukocytes and buccal mucosae cells. This phenomenon may participate in the occurrence of increased incidence of cancer in older age and on relatively frequent SMNs. The treatment of normal human T lymphocytes and fibroblasts with doxorubicin or etoposide in vitro induced significant shortening of telomeres, decreased telomerase activity, and diminished expression of telomerase reverse transcriptase (TERT) and telomere binding proteins TPP1 and POT1 [70]. Several studies described that a shortening of peripheral blood leukocytes telomeres was detected in survivors of familiar and sporadic breast cancer, childhood leukemia, spinocellular head and neck cancer, and non-Hodgkin’s lymphoma [71,72,73,74]. Telomeres examined in peripheral blood leukocytes of survivors of different cancers are shorter than telomeres of age-matched controls, and this shortening is accompanied by chromosomal aberrations [75,76]. Lee et al. suggest that in non-Hodgkin’s lymphoma patients, hematopoietic stem cells lose telomere length during the recovery period from bone marrow suppression after conventional-dose chemotherapy because: (i) mean telomere length in blood leukocytes was shorter after chemotherapy than before chemotherapy; (ii) mean telomeres length was shorter after chemotherapy than in age-matched healthy controls; (iii) there was no correlation between the extent of telomere shortening and time after chemotherapy [74].

Finnish studies [77,78,79] focused on late effects of therapy of high-risk neuroblastoma, which also included high dose chemotherapy followed by autologous transplantation of hematopoietic progenitor cells, reported signs of premature arterial aging (increased common carotid artery intima-media thickness, plaque formation, and decreased arterial lumen), shorter telomeres and higher serum levels of CRP. They suppose that all those findings are signs of premature aging. Fumagalli et al., in their experiments, proved that damage to telomeric DNA caused by gamma irradiation or by cytostatics is not reversible [80].

Gramatges et al. demonstrated an association between telomeres shortening in buccal cells and SMNs in childhood cancer survivors. Analysis of most common subsequent cancers (thyroid cancer, breast cancer, or sarcoma) found a statistically significant correlation only in thyroid cancer. However, significant associations could not be demonstrated for subsequent breast cancer or sarcoma, probably because of a low number of cases [81].

Based on the above facts, we suppose that telomeres damaging caused by chemo- and/or radiotherapy may be one of the mechanisms of anticancer therapy late effects, including SMNs.

We suppose that immunological defects in cancer survivors could also be involved in creating SMNs because an increased incidence of tumors has been repeatedly reported in immunocompromised individuals. This also assumed by Chattopadhyay et al., in their study based on the analysis of non-Hodgkin’s lymphoma as an SMN, that immunosuppression is a crucial mechanism for forming SMNs [82].

We investigated the frequency of lymphocyte populations in a group of 229 Hodgkin’s lymphoma long-term survivors. The most frequent pathological findings were decreased CD3+ and CD4+ proportions and a particularly low CD4/CD8 ratio. Those changes were more frequently found in the group with recurrent infections [2]. On the contrary, these changes were not common in nephroblastoma survivors. This may be due to different treatment protocols used for these two cancers and genetic influences because, even in nephroblastoma survivors treated with radiotherapy, these changes of T lymphocytes subpopulations were rare [3]. Cimino et al. also evidence the importance of genetic influences for immune disorders in Hodgkin’s lymphoma survivors. They found in long-term survivors of Hodgkin’s lymphoma from families with multiple cases at any age also decreased CD4/CD8 ratio and lower response to the polyclonal mitogens PHA and Con-A. In healthy relatives from these families, the immunological findings were similar to those in survivors [83]. In contrast, high-risk neuroblastoma survivors after very intensive therapy in the first years after finishing therapy had changes in the frequencies of CD8 cell subpopulations, but five years after treatment, the subpopulations were normalized [84]. Daniel et al., in their study, detected changes in immune functions in CCSs, particularly in those who underwent total body irradiation with hematopoietic stem cell transplantation. Those changes were accompanied by accelerated epigenetic aging of T cells [85].

Repeated courses of chemotherapy, antibiotics, and pneumocystis prophylaxis administered in leukemia treatment causes changes in the microbiota that never recover and contribute to several illnesses, including SMNs predisposition. Changes were found in microbial composition in ALL CCSs, who were many years post-treatment, compared with their siblings [86]; indeed, this supports the long-lasting effects of treatment on the gut microbiome. Another study correlated microbiota composition with inflammatory and T lymphocytes markers in long-term ALL CCSs and healthy controls, where they found enrichment of Actinobacteria and depletion of Faecalibacterium in survivors. Those microbiota changes correlated with increased plasma concentrations of IL-6 and CRP and HLA-DR+CD4+ and HLA-DR+CD8+ T cells, indicating inflammation and immune activation [87]. Because bacteria, viruses, and fungi are associated with tumorigenesis by affecting the immune system, inflammation, cellular signaling, and cell energetics [88], it can be hypothesized that the above changes in the microbiome could contribute to the formation of SMNs.

It follows from the above that chemotherapy plays an important role in the development of SMNs, in which several different mechanisms may be involved.

## 5. Radiotherapy

Ionizing radiation has been shown to be carcinogenic, as evidenced by data from survivors of the atomic bomb, nuclear power plant disasters, and elevated cancer risk in occupational radiation exposure [89,90]. However, radiotherapy is a standard part of the treatment of many cancers, including childhood.

Several models of radiation dose-response of SMNs have been published, but this relationship is not proved in all SMNs. More information is available for adult tumor survivors since there are larger cohorts of survivors. Berrington De Gonzalez et al. analyzed the results of 28 studies focused on the incidence of SMNs in survivors after radiotherapy, and found a downturn in the dose-response curve for radiation-related SMNs only in thyroid cancer, even for doses of >60 Gy. For thyroid cancer, there was a plateau and then a decrease in risk at doses >20 Gy in some studies [91]. The risk of radiation-induced cancer was significantly smaller after radiotherapy than in the Japanese atomic bomb survivors, which the authors explain by uniform dose adjustment and dose fractionation in therapeutic irradiation [91]. Sachs et al. proved, by analysis of SMNs, that the risk of secondary cancer does not decrease at high doses, contrary to the predictions. It was previously thought that SMNs induction could be reduced at high doses due to cell killing; the result would be the so-called “inverted U curve”. Sachs’ model supposed that stem cell repopulation neutralizes radiation-induced cell killing, so the risk is approximately linear [92]. Another theoretical model that extended the former model based on cell mutations also includes radiation-induced inflammation and proliferative stress [93].

A large study of survivors of adult cancers, on whom radiotherapy was used, found that only a small proportion of SMNs are caused by radiotherapy in adult cancer survivors [94] as opposed in CCSs [95,96,97]. The risk of developing SMNs depends not only on the dose of radiation but also on the extent of the irradiated field, the radiation technique, on genetic influences (as mentioned above), the associated chemotherapy, and the age of irradiation [89]. The combination of chemotherapy and radiotherapy increases the risk of developing SMNs almost twofold compared to the risk after chemotherapy alone or radiotherapy alone [12].

Almost all studies focused on the occurrence of SMNs in CCSs demonstrate radiotherapy as a significant risk factor. Thyroid carcinoma after irradiation of the neck, breast cancer in the female after irradiation of the chest, gliomas and meningiomas after neurocranial irradiation, soft tissues and bone sarcomas, salivary gland cancers, and NMSC in the irradiated field are most often described [95,96,97]. There has also been an increased incidence of subsequent leukemia, notably AML and MDS, after radiotherapy [33,56]. The leukemogenic effects of radiation are supported by the increased incidence of myeloid malignancies in survivors of the atomic bomb explosions, with the highest incidence at 5–7 years after exposure [56].

Radiotherapy techniques have made considerable progress in recent years, so it is hoped that patients currently treated with radiotherapy will be at less risk of developing SNs [89].

## 6. Other Therapeutic Modalities

Recently, there have been first reports of the occurrence of SNs after targeted treatment and immune checkpoint inhibitors in adult tumor survivors. RAF inhibitor Vemurafenib induces skin toxicity, including induction of squamous carcinoma [98]. There was a described occurrence of hematophagous histiocytosis after therapy by blinatumomab [99]. Adult cancer survivors treated by anti-PD-1/PD-L1 therapy had an occurrence of some SMNs, but it is difficult to judge whether immune checkpoint inhibitors were involved [100]. There is still little information on the late effects, including SMNs of targeted treatment and immune checkpoint inhibitor therapy, even in adults, but more information is likely to emerge as this treatment expands.

There is no mention in the literature of the importance of surgical treatment for the development of SNs, and even splenectomy does not increase the risk of SNs [23,101,102,103]. However, some studies report a higher incidence of secondary leukemias in Hodgkin’s lymphoma survivors who have been splenectomized [104,105,106,107], and splenectomy in Hodgkin’s lymphoma is currently only rarely used.

## 7. Conclusions and Future Directions

The risk of SNs in CCSs is probably mainly influenced by the type of first cancer, the length of survival, the therapy used, and genetic factors. In cancers with a very good prognosis, such as childhood Hodgkin’s lymphoma, nephroblastoma of a favorable histological type or seminoma in young adults, the late effects, especially SMN’s development, significantly affect long-term survival. Therefore, there is an effort to reduce particularly hazardous treatments such as radiotherapy. In current protocols for therapy of Hodgkin’s lymphoma, children with negative PET scans after chemotherapy do not receive radiotherapy. In nephroblastoma, radiotherapy is dictated by stage, histological type, and response to preoperative treatment, and only a low number of them require radiotherapy.

Current recommendations for SNs surveillance are based upon known clinical risk factors, such as patient demographics and therapeutic exposures (radiation, drugs). Methods for predicting risk for SNs in childhood cancer survivors have been proposed based upon statistical modeling that incorporates clinical and demographic variables [108]. Incorporation of other laboratory parameters into such predictive algorithms would be helpful in SNs surveillance; for personalized surveillance for the risk of SNs, data are necessary from large long-term follow-up studies of CCSs that combine clinical and laboratory parameters. The first consensus recommendations for thyroid cancer surveillance in CCSs and AYA cancer survivors, which takes into account many of factors (radiation dose rate, fraction size, age, gender, thyrotrophin levels, concurrent chemotherapy, genetic susceptibility (family history of thyroid cancer)), has already been prepared [109]. One may also speculate that in the future it will be possible to therapeutically affect chemotherapy-induced telomere shortening. A phase 1/2 prospective study involving patients with telomere diseases found that treatment with danazol led to telomere elongation [110].

Increased attention should be paid to genetic syndromes. In our opinion, genetic consultation should be offered to all survivors with SNs. Further research should also focus on finding genetic risk factors for the development of SNs, such as polymorphisms. In the future, it is possible to assume individualized treatment that is also based on the risk of developing SMNs, e.g., the omission of a drug in polymorphism with increased risk of SMNs.

## Figures and Tables

**Table 1 cancers-13-05064-t001:** Childhood and young adults cancer predisposition syndromes [30,31,35] (https://omim.org, accessed on 15 May 2021; https://www.orpha.net; https://rarediseases.org; https://www.ncbi.nlm.nih.gov, accessed on 15 May 2021).

Genetic Syndrome	Gene(s)	Associated Cancer
WAGR	*WT1, PAX6*	nephroblastoma
Beckwith- Wiedemann	*CDKN1C, H19, KCNQ1*, *NSD*	nephroblastoma, hepatoblastoma, adrenocortical, embryonal rhabdomyosarcoma
Denys-Drash	*WT1*	nephroblastoma, gonadoblastoma
Perlman	*DIS3L2*	nephroblastoma frequently bilateral
Hereditary retinoblastoma	*Rb1*	retinoblastoma frequently bilateral, osteosarcoma, melanoma
Ataxia-Telangiectasia	*ATM*	leukemia, lymphoma, medulloblastoma, glioma, skin, stomach, ovarian, breast, thyroid, uterine
Bloom	*BLM*	leukemia, lymphoma, nephroblastoma, osteosarcoma, head and neck, lung, esophageal, breast, skin
Fanconi anemia	*FANCA*, *FANCB*, *FANCC*, *FANCD1*, *FANCD2*, *FANCE*, *FANCF*, *FANCG*, *FANCI*, *FANCJ*, *FANCL*, *FANCM, FANCN*	AML, MDS, hepatocellular, nephroblastoma, neuroblastoma, head and neck, esophageal, breast, cervical, vulval
Nijmegen breakage	*NBS1* rarely *LIG4*	leukemia, lymphoma, medulloblastoma, rhabdomyosarcoma
Xeroderma pigmentosum	*XPA*, *ERCC3*, *XPC*, *ERCC2*, *XPE*, *ERCC4*, *ERCC5*, *POLH*	NMSC, melanoma, medulloblastoma, glioblastoma, lips, mouth and tip of tongue cancer, leukemia, colorectal and lung
Werner	*WRN*	soft-tissue sarcoma, melanoma, osteosarcoma, thyroid, leukemia, lymphoma, meningioma
Neurofibromatosis I	*NF1*	glioma, gastrointestinal stromal tumor, dermal neurofibroma, malignant peripheral nerve sheath tumor, leukemia, rhabdomyosarcoma, neuroblastoma, pheochromocytoma
Neurofibromatosis II	*NF2*	schwannomas, meningioma, ependymoma, low grade gliomas
Legius	*SPRED1*	lipoma, desmoid, breast, glioma
Noonan	*PTPN11*, *SOS1*, *RAF1*, *RIT1*, *KRAS*	neuroblastoma, leukemia, glioma
Costello	*HRAS*	papilloma, neuroblastoma, bladder, embryonal rhabdomyosarcoma
Bohring-Opitz	*ASXL1*	medulloblastoma, nephroblastoma
Mulibrey nanism	*TRIM37*	thyroid, ovarian, renal papillary and endometrial, nephroblastoma, pheochromocytoma, leukemia
Simpson-Golabi-Behmel	*GPC3* or *GPC4*	medulloblastoma, nephroblastoma, neuroblastoma, hepatoblastoma, gonadoblastoma
Familial Adenomatous Polyposis	*APC*	colorectal, stomach, small intestine, pancreatic, thyroid, cholangiocarcinoma, medulloblastoma, hepatoblastoma, desmoids
MUTYH- polyposis	*MUTYH*	colorectal, duodenal, thyroid, ovaries, bladder, skin
Peutz Jehgers	*STK11*	colorectal, gastric, breast, lung, pancreatic, uterine, ovarian, testicular
Juvenille polyposis	*SMAD4*, *BMPR1A*	colorectal, stomach, pancreatic
Von Hippel Lindau	*VHL*	pheochromocytoma, pancreatic neuroendocrine tumors, renal, hemangioblastomas
Hereditary paraganglioma/pheochromocytoma	*SDHA*, *SDHB*, *SDHC*, *SDHD*, *SDHAF2*, *TMEM127*, *MAX*	paraganglioma, pheochromocytoma, gastrointestinal stromal tumor
Multiple Endocrine Neoplasia 1	*MEN1*	pituitary, parathyroid adenoma, ependymoma, meningioma, neuroendocrine pancreatic
Multiple Endocrine Neoplasia 2A/2B	*RET*	neuroendocrine tumor, adrenal adenoma, insulinoma, medullary thyroid, pheochromocytoma
Hyperparathyroid- Jaw tumor	*CDC73*	parathyroid, jaw ossifying fibroma, nephroblastoma, renal, uterine, ovarian, testicular, thyroid, pancreatic
Rhabdoid tumor predisposition	*SMARCB1*, *SMARCA4*	atypical teratoid/rhabdoid tumor, schwannoma, meningioma, malignant rhabdoid tumor, ovary
Frasier	*WT* 1 intron 9	gonadoblastoma
Gorlin	*PTCH1*, *SUFU*, *PTCH2*	basal cell, medulloblastoma SHH group, meningioma, fibrosarcoma, nephroblastoma, rhabdomyosarcoma
PTEN hamartoma tumor	*PTEN*	breast, thyroid, renal, colorectal, endometrial, melanoma
PROS	*PIK3CA*	nephroblastoma
Tuberous sclerosis complex	*TSC1*, *TSC2*	hamartomas, astrocytoma, angiomyolipoma, renal cell, neuroendocrine
Hereditary pleuropulmonary blastoma	*DICER1*	pleuropulmonary blastoma, pineoblastoma, meduloepithelioma, thyroid, cystic nephroma, renal sarcoma, nephroblastoma, mesenchymal hamartoma; ovarian, rhabdomyosarcoma
Dyskaratosis congenita	*TERT*, *TERC*, *DKC1*, *TINF2*	squamous cell- head and neck, anus, skin, gastric, MDS, leukemia
Rothmund-Thompson	*RecQL4*	osteosarcoma, skin
Familial atypical multiple mole melanoma	*CDKN2A*	astrocytoma, melanoma, pancreatic cancer
Li-Fraumeny	*TP53*	sarcoma, breast, brain, adrenal glands
Schwannomatosis	*SMARCB1*, *LZTR1*	schwannoma
meningioma predisposition	*SMARCE1*	meningioma
Lynch syndrome	*MSH2*, *MSH6*, *MLH1*, *PMS2*, *EPCAM*	colorectal, stomach, small intestine, liver, gallbladder ducts, urinary tract, brain, skin
MEN4	*CDKN1B*	similar to MEN1
Familial thyroid cancer	*RET*, *NTRK1*	thyroid
Sotos	*NSD1*	leukemia, lymphoma, nephroblastoma, hepatocarcinoma, neuroblastoma
Rubenstein–Taybi	*CREBBP*, *EP300*	meduloblastoma, oligodendroglioma, neuroblastoma, meningioma, rhabdomyosarcoma pheochromocytoma
Schinzel–Giedion	*SETBP1*	malignant sacrococcygeal teratoma, hepatoblastoma, primitive neuroectodermal tumor
NKX2-1	*NKX2-1*	nonmedullary thyroid
Hereditary leiomyomatosis and renal cancer	*FH*	leiomyoma, renal carcinoma, pheochromocytoma
Metabolic disorders	*L2HGA*, *FAH*	brain tumors (anaplastic ependymoma, low grade glioma, meduloblastoma, glioblastoma)
Turcot	*APC*, *MLH1*, *MHS6*, *MSH2*, *PMS2*	colorectal, brain
Gardner	*APC*	colorectal, desmoid, osteoma

**Table 2 cancers-13-05064-t002:** Characteristics of secondary AML [57,61].

	Topoisomerase II Inhibitors	Alkylating Agents
Interval from treatment	1–3 y	5–7 y
FAB classification	M4/M5	M1/M2
Karyotype abnormalities	involving *MLL* at 11q23	(−5)/del(5q), (−7)/del(7q)
Preceding MDS	Rare	yes
Age association	Young	older

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
