# Peer review of "Possible Mechanisms of Subsequent Neoplasia Development in Childhood Cancer Survivors: A Review"

_cancers, 2021, doi:10.3390/cancers13205064_

Round 1
Reviewer 1 Report
The text by Kruseova et al. describes the possible mechanisms underlying the development of secondary tumors after healing from a malignant tumor in the pediatric age. The authors describe the role of genetic factors but also chemotherapy, radiotherapy and new targeted drugs on the development of secondary tumors.
The text does not add relevant notes to the current literature. It is in fact known that chemotherapy, genetics and radiotherapy are risk factors for secondary cancers. Because of the title, the molecular structure should also be analyzed and, for this reason, my advice is to reduce the analysis to a single type of tumor.
Author Response
We thank for your review, we accepted your suggestions. We have added some data on the molecular mechanisms of SMNs development and also our new results that demonstrate coincidence of SNs and renal sporadic angiomylipomas. Language review by a native speaker was performed.
Reviewer 2 Report
General comment : This paper is an exhaustive review of the literature on risk factors for second cancers in childhood cancer survivors. It makes an interesting current point on the subject.
Minors comments :
Title : you should indicate in the title that it is a review
Page 8 line 186: “teleradiotherapy” => should rather use “external radiotherapy”
Page 13 line 388-395 This paragraph refers to adult cancer survivors which is not the subject of the paper. The 8% excess risk is for adults. I feel that this scrambles the message that should target on the risk in childhood cancer survivors. I suggest to delete this reference.
Author Response
We thank for your review, we accepted your suggestions. We have expanded the name with an appendix-Review. We used the term “external radiotherapy” instead of “teleradiotherapy”. Paragraph in lines 388-395 we significantly shortened and left the citation only to demonstrate that SMNs after radiotherapy are less common in adults than in children. We have added some new information, including our new results. Language review by a native speaker was performed.
Reviewer 3 Report
This is a nice overview of the question, which presents cmprehensively our current knowledge about this question without any striking information.
Author Response
We thank for your review. We have added some new information, including our new results. Language review by a native speaker was performed.
Round 2
Reviewer 1 Report
The paper has improved significantly since the first review.
It is a very complex problem where different factors such as genetics, chemo-radiotherapy, gender, etc. etc. play a significant role in the development of secondary tumors.
As a minor issue: The authors should also discuss about the age of the patients in correlation to the type of primary and secondary cancer, this would allow to better define the follow-up program of individual patients, regardless of any genetic syndrome.
Author Response
Thank you for your review, we supplemented the diagnosis of the first and subsequent tumors both in childhood cancer survivors and in adolescent and young adult cancer survivors.